# Magnetic Nanomaterials for Arterial Embolization and Hyperthermia of Parenchymal Organs Tumors: A Review

**DOI:** 10.3390/nano11123402

**Published:** 2021-12-15

**Authors:** Natalia E. Kazantseva, Ilona S. Smolkova, Vladimir Babayan, Jarmila Vilčáková, Petr Smolka, Petr Saha

**Affiliations:** 1Centre of Polymer Systems, Tomas Bata University in Zlín, Třída Tomáše Bati 5678, 760 01 Zlín, Czech Republic; smolkova@utb.cz (I.S.S.); babayan@utb.cz (V.B.); vilcakova@utb.cz (J.V.); smolka@utb.cz (P.S.); saha@utb.cz (P.S.); 2Polymer Centre, Faculty of Technology, Tomas Bata University in Zlín, Vavrečkova 275, 760 01 Zlín, Czech Republic

**Keywords:** magnetic hyperthermia, arterial embolization hyperthermia, magnetic nanoparticles, embolic agents, animal model, clinical application (results)

## Abstract

Magnetic hyperthermia (MH), proposed by R. K. Gilchrist in the middle of the last century as local hyperthermia, has nowadays become a recognized method for minimally invasive treatment of oncological diseases in combination with chemotherapy (ChT) and radiotherapy (RT). One type of MH is arterial embolization hyperthermia (AEH), intended for the presurgical treatment of primary inoperable and metastasized solid tumors of parenchymal organs. This method is based on hyperthermia after transcatheter arterial embolization of the tumor’s vascular system with a mixture of magnetic particles and embolic agents. An important advantage of AEH lies in the double effect of embolotherapy, which blocks blood flow in the tumor, and MH, which eradicates cancer cells. Consequently, only the tumor undergoes thermal destruction. This review introduces the progress in the development of polymeric magnetic materials for application in AEH.

## 1. Introduction

Cancer is the second leading cause of death around the world. According to the World Health Organization statistics report, 19.3 million new cancer cases were diagnosed worldwide, with almost 10 million deaths from cancer in 2020 [1]. It is also notable from this report that the number of deaths due to cancer does not change from year to year, notwithstanding the application of new drugs and combination treatments. Moreover, the number of cancer patients is expected to rise to almost 30 million annually by 2040 (Figure 1).

New approaches to cancer detection and treatment are needed. The current tendency in oncotherapy is focused on the complex palliative therapy in treating cancer patients. One of such methods is Hyperthermia (HT) combined with RT and ChT, standardized by different organizations such as Radiation Oncology Group, European Society for Hyperthermic Oncology, and others [2,3,4,5]. Hyperthermia in oncology refers to the treatment of malignant diseases by controlled heating between 39–45 °C for a period of time with minimal unwanted side effects. Depending on tumor location and tissue volume, conventional HT is subdivided into three categories: local, locoregional, and whole-body [5,6]. Local hyperthermia aims to increase the temperature of near-surface primary malignant tumors before the metastases stage by ultrasound, electroporation, and more often by converting electromagnetic energy into heat. Locoregional hyperthermia is useful for large inoperable deep-seated tumors and is based on perfusion of organ and body with heated fluids or electromagnetic energy. Whole-body hyperthermia is used for patients with solid metastatic tumors. This type of hyperthermia is based on heating the blood in extracorporeal circulation using infrared radiation, hot water blankets, or thermal chambers. The medical hyperthermia devices using electromagnetic waves are called applicators. According to the heating principle, applicators are principally divided into dielectric applicator systems (capacitive heating applicator) and inductive heating systems (inductive heating applicators). A detailed description of HT technology currently used in clinical practice is presented in a book by Andre Vander Vorst [7] and review articles by H. Petra Kok et al. [8] and H. Dobšíček Trefna et al. [9].

Hyperthermia in combination with RT and ChT is widely used in Europe, the United States, Japan, China, Russia, and other countries for the treatment of different tumor types and sites: mammary gland, prostate gland, lung, liver, intestinal tract, bonny tissue, glioblastoma, etc. [9,10,11,12,13,14,15,16,17,18,19,20,21]. Preclinical in vitro and in vivo studies have shown two aspects of cancer inhibition by combining HT with RT and ChT: the death of individual cells from hyperthermia and enhancing the effects of RT and ChT [15]. The additional benefit of HT in combination with RT and ChT has been shown in a number of randomized clinical trials. Thus, for example, the overall response rate increased from 38% to 60% for patients with breast cancer who received HT + RT, while the treatment of cervical cancer at stage IIB-III-IVA with a combination of RT, ChT, and HT showed improved complete response rates and significantly increased overall survival [16,17]. Combining these methods has also proven to be effective in the preoperative treatment of stage III lung cancer [19]. According to the results obtained, the overall response to treatment was about 94%, including a complete response of about 22% and a partial response of about 72%. The significant regression of the tumor achieved in the preoperative period makes it possible to reduce the volume of surgical treatment and thereby facilitate the course of the postoperative period.

Currently, hyperthermia’s cellular and molecular basis and its effect on cancer treatment in combination with RT and ChT are better understood due to the substantial technical improvement in the sources used to supply heat and measure its output. Depending on the applied temperature and duration of treatment, various biological effects of hyperthermia on macroscopic and microscopic levels have been revealed [18,22,23].

The dominant mechanisms of cancer cell death caused by heating tissues to a temperature within the range of 41–45 °C are necrosis, apoptosis, and modes related to mitotic catastrophe [24,25,26,27,28]. The macroscopic effect of hyperthermia manifests itself in the tumor’s vascular system, i.e., heat increases the blood flow, which increases the vessel permeability and tissue oxygenation, which, in turn, causes a temporarily increased radiosensitivity. Concerning the microscopic effect, hyperthermia causes changes in the cellular components of the tumor and thus leads to a loss of cellular homeostasis. The mechanisms involved in heat-induced cell damage are protein denaturation, lipid peroxidation, and DNA damage. Moreover, hyperthermia modulates the immune system due to the production of heat shock proteins (HSPs), which, in turn, stimulates macrophages by acting in damage-associated molecular patterns. On the other hand, HSPs protect cells from apoptosis, which reduces the effect of hyperthermia.

The chemo-sensitizing property of hyperthermia is determined by (1) an increase in the cellular membrane permeability so that chemotherapeutic drugs can more easily pass the cell barrier, (2) a drug-induced DNA adduct formation, and (3) inhibition of DNA repair, which all enhance the cytotoxic activity of drugs [14,27]. Moreover, intratumoral heat affects DNA damage pathways by deactivating specific repair proteins. These processes are highly dependent on several factors: the degree of temperature elevation, the duration of heat, and the cell type and microenvironmental conditions, such as the acidity and oxygenation status of the tumor. In addition, hyperthermic chemosensitization depends on the type and concentration of drugs due to different mechanisms by which heat affects drug activity, i.e., transport, intracellular cytotoxicity, and metabolism. In vivo and in vitro studies, as well as clinical results, have demonstrated that most chemotherapeutic drugs are effective when delivered just before or during an HT session [28].

Critical problems of conventional hyperthermia in clinical practice are insufficient heat localization in the tumor, especially in deep-seated tumors, heterogeneity in temperature distribution within the tumor, and overheating of healthy tissues due to deficiency in temperature monitoring. For these reasons, the use of hyperthermia alone gives an overall response of only about 15% [5]. The main challenge of hyperthermia is to achieve a precise energy delivery and controlled heating of primary and metastasized tumors while avoiding heating of normal tissues and overcoming the thermotolerance [26]. In particular, MH proposed in 1957 by R.K. Gilchrist as local hyperthermia [29] is still undergoing preclinical and clinical trials as an independent method for cancer therapy and as a multimodal treatment in combination with RT and ChT [30,31,32,33,34,35,36,37,38]. The general methodology of MH comprises the introduction of magnetic material into the tumor followed by exposure to an alternating magnetic field (AMF) at moderate frequencies and amplitudes (f = 0.05–1.5 MHz, H ≤ 15 kA·m^−1^) to limit peripheral nerve stimulation due to induced eddy currents occurring in the body [39,40,41]. The free current loss also needs to be considered since it may lead to nonspecific induction heating. The heating ability of magnetic material in AMF is usually estimated by specific power loss (SLP) or specific absorption rate (SAR), which are defined as heating power (P [W]) generated per unit mass of magnetic nanoparticles (*m*_MNP_ [g]): SAR = P/*m*_MNP_ [42,43]. The heating power produced by the mediator depends on nanoparticles (NPs) concentration, core size, and magnetic properties (saturation magnetization, magnetic anisotropy energy), the viscosity and heat capacity of dispersion media, as well as on the extrinsic factors, i.e., frequency and amplitude of AMF [7]. To eliminate these extrinsic factors, intrinsic loss power (ILP) as a system-independent parameter was introduced to compare results obtained in different field conditions: ILP = SAR/f × H^2^ [44].

The key requirement in MH is maximizing heat generation within medically safe limits of the AMF. Therefore, particle size and particle size distribution must be taken under control. Experimental determination of the heating effect of magnetic materials is usually conducted by the nonadiabatic calorimetric method [45] and rarely by adiabatic calorimetry [46]. It is also possible to predict the size-dependent heating efficiency of MNPs by stochastic Neel–Brown Langevin equation and Monte Carlo (MC) simulations [47]. U.M. Endelmann et al. used this method to calculate the heating efficiency of MNPs with a size of 10–30 nm and various values of effective anisotropy constant (K = 4000 J/m^3^–11 J/m^3^) and damping parameter (α = 0.5–1). The magnetic parameters of MNPs at the same time were obtained using VSM. Experimental and simulated results have shown that the maximum SLP value demonstrated particles in the 22–28 nm range. Moreover, MC simulation revealed a strong dependence of SLP on K [48]. Besides, various empirical and analytical methods are used to evaluate the SLP from an experimental setup, such as the initial slope, corrected slope, Box–Lucas, and steady-state methods [49]. Although recently developed bioheat models for MH are used to understand heat transfer phenomena in living tissue. These methods are fully considered in newly published articles by I. Raouf et al. [50] and M. Suleman et al. [51]. 

To optimize the experimental conditions, magnetic fluids of different compositions and volumes were analyzed in several laboratories. As a result, the necessary AMF parameters and sample volume were determined as f = 300 kHz; H = 10.6 kA·m^−1^–15 kA·m^−1^, volume −1 mL [52].

For MH, the concentration, distribution, and retention of magnetic material within tumor volume are critical parameters. Currently, there are two main directions in MH dependent on the magnetic heating agent used and the manner of its intratumoral administration. Those are «Magnetic Fluid Hyperthermia» (MFH) [41,42,43,44,53,54] and «Arterial Embolization Hyperthermia» (AEH) [55,56,57,58,59,60,61]. These methods are based on the use of a liquid carrier medium typically containing magnetic iron oxide nanoparticles due to their good biocompatibility. The carrier medium is usually water or saline in MFH, while, in AEH, it is oily contrast media (Lipiodol Ultra Fluid, France, and its analogs) [56,57,58,59,60,61,62,63], and in-situ gelling materials [64] and low-viscosity polymers [65,66,67,68,69].

In clinical practice, the main problem of MFH is the way mediator administration which is realized either by direct intratumoral injection or intravenous medication that do not provide a uniform distribution of magnetic phase in the target tissue due to the typical heterogeneous structure of malignant tumors [54]. Therefore, heat distribution within tumors is not uniform, and there is a risk of proliferation of survived cancer cells. This method is also unacceptable for treating patients with hepatocellular carcinoma (HCC) due to the bleeding tendency in such highly vascularized tumors. Another problem of MFH is the low accumulation of particles in the cancerous area due to metabolism. Based on the clinical trials, the dose of magnetic phase for effective MFH is reported to be 40 mgFe/mL_tissue_ per site, which is difficult to achieve in practice [31]. Indeed, as S. Wilhelm et al. reported, only 0.7% of nanoparticle dose can be delivered to a solid tumor [59]. However, despite these challenges, MFH has now received approval for clinical testing in humans by the German Federal Institute for Drugs and Medical Devices and the United States Food and Drug Administration to treat glioblastoma and prostate gland by colloidal suspension of aminosilane-coated iron oxide nanoparticles (NanoTherm^®^, MagForce company, Berlin, Germany) [70,71,72]. This is most likely due to the development of a unique inductive heating applicator operating at an AMF frequency of 100 kHz with a field strength of up 18 kA/m (MFH 300F NanoActivator^®^; MagForce Nanotechnologies AG, Berlin, Germany) [30].

In contrast to MFH, the challenges discussed above can be eliminated in AEH, developed to treat parenchymal organs. The concept of AEH is based on selective capacitive or inductive hyperthermia after transcatheter embolization of a tumor’s arterial supply with a mixture of magnetic particles and an embolic agent [65,66,67,68,69]. As a result of embolization, the size of the tumor decreases due to a decrease in the blood supply, leading to partial necrosis of the tumor. The technique for transarterial embolization dictates the choice of materials for AEH. It depends on the patient’s structural features and the treated lesion. However, the technique should meet the requirements of nontoxicity, nonantigenicity, stability to lysis, and radio-opacity. Moreover, at the delivery stage, the material should be of low viscosity to pass through angiographic catheters and fill up not only the main artery but also peripheral arteries and small blood vessels, i.e., ensure both proximal and distal embolization [66,67]. Then, the material should prevent blood flow, for example, due to the rapid increase of the material’s viscosity.

The combined effect of embolization and hyperthermia on the tumor leads to ischemic necrosis of the tumor and programmed cell death, apoptosis. The first clinical trials of AEH were conducted 20 years ago at the Russian Research Centre for Radiology and Surgical Technologies (St. Petersburg, Russia) with the permission of the Russian Ministry of Health [67]. At the first stage of treatment, X-ray endovascular embolization with silicone composition containing microsized carbonyl iron particles (Ferrocomposite^®^, Linorm, Saint Petersburg, Russia) was performed for 46 patients with stage III and IV renal cell carcinoma of the kidney [66]. The occlusion of the vascular system of the kidney was controlled angiographically. Then, 7–10 days after the post-embolization period, capacitive RF hyperthermia was performed at a frequency of 27.12 MHz with an input power of 80 W. The temperature at the treatment area was monitored by an invasive method: needle-shaped temperature sensors were inserted into the peripheral parts of the kidney under ultrasound control. The treatment time of 30–45 min was necessary to heat a tumor to 43–45 °C. Morphological and histological analysis of kidneys after palliative nephrectomy showed the complete occlusion of the renal tumor blood supply and massive necrosis of the tumor tissue (Figure 2) [66]. As a result, 3–5-year survival of inoperable patients after embolization and hyperthermia was about 18% and 5%, respectively. Nevertheless, AEH based on the dielectric heating principle remains an experimental method in medical practice due to the difficulties associated with the overheating of healthy tissue. Contrariwise, the AEH method involving inductive heating principle (induction hyperthermia) has the advantage to heat selectively a tumor filled with ferromagnetic material without heat generation in the fat layers [68,69,73,74,75].

The purpose of this review is to demonstrate the potential of AEH for the treatment of deep-sited tumors of the vascular organs, kidneys, liver, and pancreas gland. Considering the strict limitations on the frequency and amplitude of AMF in MH, the heating ability of the mediator should be maximized considering the physical mechanisms responsible for the losses in magnetic materials. Therefore, the mechanisms of magnetic losses in nanomaterials are discussed to determine the relationship between the heating efficiency and magnetostructural properties of NPs. Besides, the role of interparticle magnetic interactions and the properties of the carrier medium on the heating efficiency are considered. The review also presents the results of in vitro and in vivo preclinical trials of magnetic nanomaterials in treating several oncological diseases using AEH.

## 2. Properties of Magnetic Materials for Their Application in Magnetic Hyperthermia: Nanomagnetism over Micromagnetism

The primary tenet of micromagnetism is that ferro and ferrimagnetic materials are mesoscopic continuous media where atomic-scale structure can be ignored since the magnetization (M), and the demagnetizing field (H_d_) are nonuniform but continuously varying functions of distance (r) [76]. The main characteristic of macroscopic samples is the irreversible nonlinear response of M when exposed to an external magnetic field (H) (Figure 3).

In irreversible processes, energy is dissipated in the crystal lattice in the form of heat, known as hysteresis loss. The actual physical processes by which energy is dissipated during a quasistatic traversal of the hysteresis loop are identical to those responsible for the dynamic losses. In most materials with multidomain structures, the hysteresis of magnetization arises from domain wall motion or domain nucleation and growth.

There are several contributions to the free energy of magnetic samples with multidomain structure [77]:

Magnetostatic energy E_m_, resulting from the interaction of atomic magnetic moments with local internal magnetic field H_i_:(1)$$E_{m}=-\int_{\mathrm{vol}}M\cdot H_{i}\;\mathrm{dV}$$

Magnetic free energy, determined by an interaction between atomic magnetic moments and crystalline lattice expressed by magneto-crystalline energy E_k_ and magnetostrictive energy E_λ_:(2)$$E_{k}=-\int_{\mathrm{vol}}f_{k}\;\mathrm{dV}$$
where f_k_ is magnetocrystalline anisotropy density
(3)$$E_{\lambda }=-\int_{\mathrm{vol}}f_{\lambda }\;\mathrm{dV}$$
where f_λ_ is magnetostriction anisotropy density.

Free energy, which is related to the magnetic-exchange interaction.

The magnetostatic energy with dipole–dipole nature is inversely proportional to the volume of the particle, while the domain-wall energy is proportional to the area of the wall (Figure 4) [78]. Considering the balance between the magnetostatic energy and the domain-walls energy, the formation of a multidomain structure is energetically unfavorable when the particle size is less than the width of the domain walls.

Modern methods for studying the micromagnetic structure of materials, such as transmission microscopy and off-axis electron holography, as well as numerical micromagnetic simulation, revealed three typical magnetic configurations (states) in magnetic materials, depending on the particle size: single-domain (SD) with a uniform arrangement of magnetic moments, pseudo-single-domain (PSD) with a vortex spin arrangement, and multidomain (MD) structure where magnetic structure breaks up into discrete regions separated by domain walls (Figure 5) [79,80,81,82].

In the case of SD particles, these configurations may exhibit two states: (1) superparamagnetic (SPM) with unstable behavior due to the thermal instability of the magnetization if the thermal energy k_B_T (k_B_ = 1.38 × 10^−23^ J·K^−1^ is Boltzmann constant, and T is temperature sufficient to change the orientation of the magnetic moment of particle, and, (2) stable SD with ferromagnetic-like behavior when the magnetic moment is pinned along the magnetic anisotropy axis as a result of effective magnetic anisotropy [83].

For many practical applications, such as magnetic storage media and MH, the suitable particle size is within the range of a stable SD state, namely, in the vicinity of SD to PSD transition where the coercivity approaches maximum (Figure 5). The critical size for an SD magnetic state depends on several parameters, including M_S_ and K. For magnetite and maghemite approved for biomedical applications, the particle size range for the stable SD state is about 20–80 nm for spherical particles and about 200 nm for elongated particles with 2:1 axial ratio [83,84].

The magnetization reversal mechanism in nanosized magnetic materials differs from that for MD ferromagnets. In SD particles smaller than 100 nm, magnetization occurs only by coherent rotation of all atomic magnetic moments within the sample against an energy barrier (ΔE) given mainly by the shape and the crystalline anisotropy fields [85,86,87,88,89,90]:(4)$$\Delta E=K-{\mathrm{HM}}_{S}\left|\sin\varphi \right|\pm {\mathrm{HM}}_{S}\cos\varphi$$
where K is the anisotropy energy density, and φ is the angle between the easy axis and the magnetic field.

The shape anisotropy comes from the demagnetizing field:(5)$$H_{d}=-N_{d}M_{S}$$
where $N_{d}$ is the demagnetizing shape factor of a magnetized unit.

A dominant effect of the size and shape anisotropy on H_C_ and M_S_ and the heating efficiency has been observed in anisotropic magnetite NPs, such as wire, ring, rod, cube, octahedron, etc. (Figure 6) [87,88,89,90].

It is known that MNPs with particle sizes larger than 20 nm are in a stable-domain state with ferromagnetic-like behavior when the magnetic moment is pinned along the magnetic anisotropy axis as a result of effective magnetic anisotropy. Such NPs exhibit much higher heat loss in AMF. This class of magnetic nanomaterials also includes the novel octahedral monocrystalline magnetite NPs obtained by thermal decomposition [91]. Owing to the octahedral morphology, these NPs show one of the largest SARs rates reported to date for a colloidal suspension of magnetite: 1000 W/gFe_3_O_4_ at 40 mT and 300 kHz. Such behavior has been explained by the shape of NPs that imprints a biaxial or bi-stable character to the magnetic anisotropy.

Besides the size and shape, other properties of magnetic NPs should be considered for modulating the heat generation, such as particle size distribution and the presence of interparticle interactions.

The polydisperse sample represents an ensemble (mixture) of particles with various magnetization states corresponding to the distributions in magnetic properties, especially the magnetic anisotropy, which governs the height of the energy barrier. Many works have been focused on how polydispersity influences the hyperthermia performance of magnetic NPs. The detrimental influence of size polydispersity (σ) on the heat outcome is usually reported as follows: heat generation can drop between 30% to 50% for σ varying between 0.2 and 0.4 [92,93,94]. However, the decrease in the heating efficiency in polydisperse materials can be associated not only with polydispersity per se but also with low (unsaturated) magnetic field strength [95]. The use of low amplitudes in the experiments results from a number of unsuccessful attempts in clinical trials to apply the amplitudes of AMF beyond 15 kA/m. For example, M. Johannsen et al. reported that patients undergoing thermotherapy treatment of prostate cancer with exposure to AMF of 100 kHz felt discomfort at amplitudes higher than 5 kA/m [30], whereas for the treatment of brain tumor, field strength up to 13.5 kA/m was reported to be well tolerated [71]. Therefore, in each treatment case, the frequency and amplitude of AMF must be correctly selected regardless of the polydispersity of particles.

As mentioned above, along with polydispersity, the interparticle interactions can significantly affect the magnetization dynamics of NPs, since they lead to aggregation, especially when particles are without surface coating.

In an ensemble of noninteracting SD NPs, the energy losses are associated with the Neel–Brown relaxation process [95]:(6)$$\tau_{N}=\tau_{0}\exp\left\{\frac{{\mathrm{KV}}_{C}}{k_{B}T}{\left(1-\frac{H}{H_{A}}\right)}^{2}\right\};\;\tau_{B}=\frac{3V_{h}\eta }{k_{B}T}$$
where τ_N_ and τ_B_ are time scales of Neel and Brownian relaxation, τ_0_ is a pre-exponential factor (10^−9^ ÷ 10^−11^ s), H_A_ is the anisotropy field equal to 2 K/µ_0_M_S_, V is the particle volume, η is the viscosity of the carrier medium, V_h_ is the hydrodynamic volume of the particle, and H is the field amplitude.

For a material in which both Neel and Brown relaxation takes place with an effective relaxation time τ_eff_, the mechanism with shortest τ_eff_ dominates. However, due to the exponential dependence of τ_N_ on the particle volume, while τ_B_ linear grows with hydrodynamic volume, different Neel and Brownian contributions can be realized for the same magnetic material with different particle sizes (Figure 7a) [95,96]. Both relaxation processes strongly depend on the amplitude of AMF. It is established that the Neel mechanism manifests itself predominantly at high field amplitudes, while at low field amplitudes, the Brownian mechanism prevails [97]. In addition, the relaxation time of a Brownian process is proportional to the viscosity of the carrier medium; thus, Brownian relaxation is largely suppressed when the particles are immobilized in a viscous medium such as cancerous tissue (Figure 7b) [98].

The assembly process of magnetic NPs in liquid media is driven by the attractive–repulsive interactions between NPs, van der Waals (vdW), magnetic, electrostatic, and solvophobic forces [98,99,100]. The former two are core–core interactions that dominate the interaction potential and hold NPs together. The van der Waals interactions scale linearly with the particle’s radius, while the magnetic interaction scales with its volume. Magnetic interactions always coexist with vdW forces, which becomes increasingly important with decreasing particle size. For example, the formation of aggregates from NPs in the absence of an external magnetic field already takes place at the beginning of coprecipitation reaction; thus, the contribution of vdW forces can be notable (Figure 8) [93]. However, the formation of dense aggregates that are stable against segregation into individual nanoparticles is possible only under the influence of interparticle magnetic interactions [100]. Estimating the threshold sizes for the agglomeration of magnetite NPs has shown that they are relatively stable against agglomeration up to 20–25 nm in diameter [101,102].

The interactions between magnetic NPs are interpreted in terms of magnetodipole and exchange interactions. Exchange interaction can be neglected when interparticle spacing is of the order of 2 nm, which approximately corresponds to the distance between two NPs with a dead layer of thickness of about 1 nm [93]. In most cases, the dominant contribution to interparticle energy is a magnetodipole coupling, which increases with the volume of NPs and depends on the mutual distance between particles [103,104,105].

Dipole–dipole interactions can be either attractive (in-line dipoles) or repulsive (antiparallel aligned dipoles). The predominant type of configuration of dipoles is the antiparallel orientation of the magnetic moments of a pair of particles [93,106]. Subsequently, the pair of dipoles stick together to form larger aggregates, and, in the absence of an external field, these aggregates have closed magnetic flux with random orientation of magnetic moments of individual NPs (Figure 9).

Even though NPs in the aggregate are in the SPM state, in some cases, the material itself may demonstrate ferromagnetic-like behavior, which is evidenced by distinct sextets on the Mössbauer spectrum and blocking temperature well above room temperature on the FC/ZFC curves (Figure 10) [107].

The ferromagnetic-like behavior of such materials can be explained by the internal structure of the aggregate formed, namely, when SPM NPs are combined into a dense 3D cluster, the so-called multicore particles [105,107,108] and nanoflowers [109]. These materials can provide efficient and rapid heating in AMF at low-field regimes [110].

The experimental results and numerical simulations have shown the different effects of inter and intra-aggregate magnetodipole interactions on heat generation [93,107,108,109,110,111,112,113]. It is found that an increase in the concentration of NPs leads to a nonmonotonic behavior of SAR (SLP) with its reduction at a specific aggregate size (Figure 11).

A theoretical study of the effect of magnetodipole interactions on the heating ability of an ensemble of particles, and especially multicore particles, is challenging since it is a multiparameter task that must account for the magnetic characteristics of primary particles, as well as morphostructural properties, i.e., polydispersity, shape anisotropy, packing density of NPs in a cluster, etc. [114,115,116,117].

An increase in the heating ability is usually explained either by a change in the characteristic height of the energy barrier related to the thermal energy [92] or by a change in the magnetic state due to the collective behavior of closely spaced NPs [84,85,113,114]. In turn, a decrease in the heating ability in the ensemble of interacting NPs is explained by the disorienting effect of a random magnetic field, causing a deviation of the magnetic moments of NPs from the direction of the AMF [116].

The nanoparticle size is one of the most important parameters that affects the magnetic properties of multicores. The study of the heating ability of multicore particles of approximately the same hydrodynamic diameter (100 nm) but formed by magnetite NPs of different sizes (7.1 and 11.5 nm) showed different results [118]. Both types of multicores improve their heating efficiency compared with individual NPs when exposure to AMF at f = 302 kHz and H = 15 kA/m, but multicores composed of larger NPs show two times higher values of SAR. The crucial role of core particle size in a cluster has also been established by C.H. Jonasson et al. They investigated the heating efficiency for SD particles of different sizes and multicore particles both experimentally and theoretically using dynamic Monte-Carlo simulations [119]. It was found that for a given AMF (1 MHz, 3–10 kA/m), core–core interactions can lead to a different character of ILP dependence on the core diameter (D_c_) when D_c_ is higher or lower than the size maximizing the ILP value, i.e., D_c_ = 20 nm (Figure 12).

Apart from magnetite and maghemite single phase-based systems, exchange-coupled magnetic NPs have been proposed so far as possible candidates for efficient MH. These particles have a core–shell structure with different combinations of magnetically soft and hard materials, for example, CoFe_2_O_4_@MnFe_2_O_4_, CoFe_2_O_4_@Fe_3_O_4_, Fe_3_O_4_@CoFe_2_O_4_, CoFe_2_O_4_@γFe_2_O_3_, FeO@Fe_3_O_4_, etc. [120,121,122,123]. The main idea underlying exchange-coupled magnetic NPs in MH is to increase the hysteresis losses by controlling the anisotropy constant (K) while maintaining superparamagnetism, thereby preventing aggregation and the formation of large clusters. The magnetic properties of exchange-coupled core–shell particles and their effect on SLP are dependent on the composition, as well as on the core and shell size (Figure 13). As can be seen from this figure, the SLP of core–shell NPs exhibits SLP values significantly higher than the SLP of single-phase magnetic NPs; however, the amplitude of AMF is twice the value allowed for medical application.

To sum up, dynamic magnetic properties and an increase in the heating efficiency of the mediator (for given amplitudes and frequencies of AMF) are determined by the following factors: (1) material composition and degree of crystallinity; (2) average particle size within the range of stable SD state, which is between 16–20 nm in diameter for ferrimagnetic iron oxides; (3) particle size distribution (preference to monodispersity over polydispersity), and (4) magnetodipole interaction. The first affects K and M_S_, the next two control the interaction strength of NPs, thus regulating the hydrodynamic size and internal structure of multicore particles, and the last one determines collective magnetic behavior and thus modifies the amount of heat generation during the hyperthermia session.

## 3. Magnetic Materials for Application in AEH

As already mentioned in the introduction, AEH is a multiple treatment modality involving transarterial embolization of tumors with magnetic material followed by exposure to AMF at clinically relevant frequencies and amplitudes. The embolotherapy itself is widely used in clinical practice for diagnostic (coronary angiography), preoperative management of malignant renal tumors, chemoembolization of malignant hypervascular tumors such as hepatocellular carcinoma, as well as in the treatment of aneurysms, hemorrhage, angiomyolipoma, and other conditions [63,67,124].

At the first stage of the treatment by AEH, transcatheter injection of the embolic agent is administered under an angiographic control. This procedure usually lasts about 20–25 min. During this time, the embolic agent should maintain low viscosity (η < 0.5 Pa/s) for transportation and filling of the tumor vascular system. After this induction period, the viscosity should increase rapidly with forming a soft and stable embolus, which can occlude the tumor blood vessels. The selective MH can be carried out after the passing of the post-embolization period of patients (fever, elevated white blood count, etc.), which usually takes from one to two weeks [66,67].

Embolic agents are generally classified into mechanical [56] and flow-directed agents, but only the latter is used in AEH as a carrier of magnetic particles. The list of such materials includes Lipiodol (Lipiodol Ultra Fluid, Guerbet, France) and its analogs [57,60,63,125], in-situ gelling materials (e.g., Onyx^®^ (DMSO, Acros Organics, Basel, Switzerland): ethylene-vinyl alcohol copolymer dissolved in dimethylsulfoxide, etc.) [64,126], and low-viscosity polymers (polyorganosiloxanes, radiopaque degradable polyurethane) [65,66,67,68,69,127,128,129].

Most studies on the effectiveness of AEH have been conducted in vivo in mouse and rabbit models with liver cancer disease after hepatic intra-arterial injection of magnetic NPs suspended in Lipiodol [59,60,61,62,63,130,131,132,133]. The use of Lipiodol as a carrier medium is determined by a set of its properties: radio-opacity (~48% of iodine), ability to induce plastic and transient embolization of tumor, microcirculations causing ischemic necrosis of tumor, limiting the ingress of viable cancer cells as well as their debris in the bloodstream. With the right choice of the magnetic particle size and concentration in Lipiodol, it is possible to achieve a homogeneous intratumoral distribution of the magnetic phase and, thus, to significantly increase the specific heating of the tumor, but only at field amplitudes of about 20 kA/m and higher, which is beyond the permissible limit in MH (Figure 14) [62].

Similar results were obtained earlier by Moroz et al., who reported the superiority of AEH compared with direct injection hyperthermia, studied on a model of a rabbit liver tumor [58,60,133]. It was found that after hepatic arterial embolization by maghemite NPs (100–200 nm) suspended in Lipiodol and subsequent MH (53 kHz, 30–45 kA/m), the tumor volume can be reduced by almost 94%, depending on the particle concentration and the uniformity of its distribution in the tumor.

Along with Lipiodol, other materials have been investigated as embolic agents in AEH. Exemplarily, the embolic properties and heating efficiency of organogel (Onyx^®^) containing silica microbeads filled with magnetic iron oxide NPs have been studied in nude mice carrying subcutaneous human carcinomas [64,126]. Thus, the intratumoral injection of nanocomposite followed by 20 min MH at 141 kHz resulted in extensive tumor necrosis (78%) but only for a group of animals exposed to AMF of high intensity (~12 mT) (Figure 15). At such an intensity of AMF, the tumor heats up to 44–45 °C, thus, undergoing thermal ablation. A survival study using magnetic resonance imaging has shown that 45% of the 12 mT-treated groups survived one year without any tumor recurrence.

Despite the demonstrated efficacy of AEH in vivo, some obstacles remain that limit the use of this method in clinical practice, namely, difficulty in assessing the actual temperature of tumors. It is generally clear that, for successful performance of AEH, two significant factors should be accomplished simultaneously: achieving the total embolization of the tumor vascular system and attaining hyperthermia temperatures causing ischemic necrosis of the tumor. Thus, a balance must be achieved between the mechanical properties of embolic agents comprising low initial viscosity, rapid solidification, and robust embolus stiffness. From that perspective, the biocompatible polyorganosiloxanes are promising embolic agents: (1) their viscosity and the curing rate can be regulated by their composition, (2) silicone elastomers do not display adhesion to living tissues, (3) they are soft materials and do not injure blood vessels [134]. Moreover, silicones closely adjoin the walls of blood vessels, reducing the probability of blood flow recovery. The low specific heat of silicon rubber of 1.05–1.30 J/g·K is also advantageous because it favors the distribution of the heat generated by incorporated magnetic particles [73]. The advantage of using these polymers for transcatheter embolization in patients with renal cell carcinoma was demonstrated with Ferrocomposite^®^ (Saint Petersburg, Linorm, Russia) [65,66,67,68,69]. The X-ray image (Figure 2a) shows a patient’s kidney with a large tumor uniformly filled with Ferrocomposite^®^. Complete occlusion of renal tumor blood supply results in necrosis of tumor tissue, and in combination with RF capacitive hyperthermia, it provided in massive necrosis (Figure 2b). Considering the positive results of clinical trials of Ferrocomposite^®^, further research aimed to improve its properties: reducing the viscosity of the embolic agent and increasing its heating ability in AMF permitted for medical application. As a result, maghemite-based silicone composition (Nanoembosil^®^) (Saint Petersburg, Linorm, Russia) was developed, which possesses a set of properties required for its use as a mediator in AEH: the ability for secure embolization of the tumor blood vessels, the high heating rate in AMF at moderate frequencies, and amplitudes, and radiopacity [68,69]. Nanoembosil^®^ has been tested in vitro at the Tomas Bata University in Zlin (CZ) and in vivo at Moscow’s Blokhin Russian Cancer Research Centre.

## 4. Nanoembosil^®^: Synthesis, Characterization, In Vitro and In Vivo Study

The magnetic phase of Nanoembosil^®^ is maghemite NPs prepared by annealing (6 h at 300 °C) of magnetite NPs synthesized by coprecipitation method under certain reaction conditions that guarantee the formation of monodisperse NPs with a high degree of crystallinity [107,135]. The annealing does not change the morphology of NPs but predictably decreases the M_S_ value. Notably, the original (as-prepared) and annealed samples demonstrate almost identical heating efficiency (Figure 16), which can be explained by increased effective anisotropy due to magnetodipole interactions resulting in the formation of stable multicore particles (Figure 8 and Figure 9).

The characteristics of raw materials used for the preparation of Nanoembosil^®^ are presented in Table 1 and Table 2.

The Nanoembosil^®^ is supplied in two compositions in separate containers: the first contains PVS, catalyst, maghemite NPs, and radiopaque potassium iodide (KI) (Container 1), whereas the second contains PHS, CTS, and PDMS (Container 2). Once the content of Container 2 is added into Container 1, the hydrosilylation reaction of hydro-and vinyl-functional silicone polymers starts. To monitor the polymerization process, the optimal concentration of reagents was chosen to provide low initial viscosity (0.25–0.3 Pa/s) during the induction period (20–25 min), followed by an abrupt increase in viscosity and the formation of a soft embolus (Table 3). To this end, the impact of each component on the kinetics of Nanoembosil^®^ formation was investigated through measurements of the rheological properties on a Rheometer with parallel plate geometry. Variation of rheological properties of the composite during its formation is presented in Figure 17.

As it can be seen from this figure, no changes in the elastic modulus (G′) and viscous modulus (G″) of complex shear modulus ${\;G}^{*}$ are observed at the beginning of the reaction, and the system’s viscosity stays constant, which corresponds to the induction period of the reaction. The complex shear modulus is defined as [136]:(7)$$G^{*}=\frac{\tau^{*}}{\gamma_{M}}=\sqrt[{2}]{\left({G\prime }^{2}+{G\prime \prime }^{2}\right)},$$
where τ^∗^ is the complex stress, and γ_M_ is the maximum value of fixed strain.

During the induction period, the composition is fluid, evidenced by a greater value of G″ over G′. As the reaction between vinyl- and hydro-groups starts, both G′ and G″ increase, but G′ increases faster than G″. At the end of the induction period (~20 min), there is an intersection between G′ and G″, indicating reaching the vulcanization point. Above the vulcanization point, the elastic component dominates over the viscous one, G′ > G″; the viscosity of the composition rapidly increases till it reaches a maximum and then stays constant after all the hydro-groups have reacted. Thus, the dominant influence on the kinetics of composite formation is exerted by PHS, PVS, PDMS, and CTS, where the last two components play the role of plasticizers to adjust composite viscosity. As to the influence of the magnetic filler concentration on the rheological properties of the composite, it is negligible up to 14 wt.%.

The heating efficiency of Nanoembosil^®^ was estimated in the AMFs at frequencies and amplitudes (f = 0.05–1.5 MHz, H ≤ 15 kA·m^−1^) standardized for medicine. The heating rate of the composites, as well as SLP, depends on the AMF parameters since the magnetization process is determined solely by the Neel relaxation (Figure 18, Table 4). Nevertheless, according to the results obtained, a high heating rate can be achieved in the entire frequency range even at sufficiently low field amplitudes. Therefore, an increase in SLP in composites of this type is possible with increased field amplitude to the allowed power level.

Considering that the embolic agent should also possess thermal expansion similar to or higher than blood to prevent the blood flow recovery during hyperthermia session, the thermomechanical properties of Nanoembosil^®^ were studied by dynamic mechanical analysis. The study revealed that the material possesses the rubber-elastic properties: shear modulus, G′, is almost independent of the applied frequency, and the loss tangent, tan δ, is slight (0.1–0.2) (Figure 19). The value of G′ within the range of hyperthermia temperatures slightly increases from 9.6 to 9.9 kPa at 38 °C and 10 Hz shear rate. The obtained shear modulus values are smaller than those reported for the artery, 30–3000 kPa [136,137]; thus, the composite will easily deform with the artery. Furthermore, the thermal expansion coefficient of the composite is 760 ppm °C^−1^ at 37 °C and slightly decreases to 710 ppm °C^−1^ with the temperature rise to 45 °C. The high value of α for the composite ensures the prevention of blood flow recovery during heating.

In vivo study of Nanoembosil^®^ was done at the N.N. Blokhin Russian Cancer Research Centre in Moscow. The study aimed to determine the embolic agent dose for intra-arterial administration and filling the tumor vascular system and estimate its antitumor effect [138]. All experiments were conducted according to the National and European guidelines on the ethical use of animals [139,140]. Throughout the procedures, animals were anesthetized with Zoletil 100 (Virbac, Carros, France). The experimental study was performed using 25 rats (8-week-old males of body weight 200 g) and 16 male rabbits weighing 2.0–3.0 kg. Some animals were kept healthy to estimate tolerability of embolization, and others were intramuscularly implanted with hepatocellular carcinoma PC-1 (rats) and VX2 (rabbits). It was established that intravenous tissue tolerance of embolic agents for rats is 0.1 mL for the composition I with η*_in_ = 0.3 Pa/s, and 0.2 mL for composition II with η*_in_ = 0.25 Pa/s, while for rabbits, 1.5 mL, which is well below the guidelines for the maximum intravenous injection volumes of experimental compounds in rats and rabbits [141]. Thus, the study of the effect of embolization on tumor growth was conducted using the established volumes of compositions. To this end, four independent animal groups with the same number of rats (*n* = 5) were used. Animals in group 1 were exposed to embolization by composition I (0.1 mL) and in groups 2 and 3, respectively, by composition II with volumes of 0.1 mL and 0.2 mL. To monitor the treatment of tumor growth, the test group received only a physiological solution.

The embolization was carried out on the 20th day after transplantation when a full regional blood flow in the tumor node was formed. The embolic agents were prepared extempore and administered once by transarterial infusion into the femoral artery using an intravenous catheter.

The efficacy of embolization was estimated according to standard criteria: inhibition of tumor growth, tumor growth dynamic (3, 7, 10, and 14 days after embolization), tumor doubling time (τ_2_), and therapeutic response (τ_trial_/τ_control_). The tumor growth was estimated by the change in the tumor volume (V_t_/V_0_), where V_0_ is the mean volume of the tumor before, and V_t_ is the mean volume of the tumor after treatment. The results of in vivo study are presented in Table 5 and Figure 20.

Accordingly, in the case of the test group of rats, the tumor growth rate increased by a factor of 1/5. The embolization of the tumor vascular system by both types of compositions significantly inhibits the tumor growth, and this effect is more pronounced in the case of embolization by Nanoembosil^®^ (composition II). Indeed, over a period of seven days after embolization, the rate of tumor growth was two times lower in group 3 compared with the test group. This is due to the low viscosity of composition II (0.25 Pa/s), which allowed one to increase the administered dose without causing side effects.

Similar results were obtained in rabbits with intramuscularly transplanted VX-2 liver cancer. Intra-arterial injection of Nanoembosil^®^ at a dose of 1.5 mL on the 20th day of tumor growth led to a significant decrease in growth rate within two weeks. Stabilization of tumor growth within 14 days is associated with tumor cytoreduction by more than 50%.

The efficacy of silicone-based magnetic composites as heat mediators was studied in vitro on the human Hepatocellular carcinoma cell line (HepG2). The treatment was conducted in the following scheme. Prior to in vitro testing, a certain amount of Nanoembosil^®^ was mixed with cells in the ratios of 1:1 and 2:1, which correspond to iron concentrations of 3.5 and 5 g/L in the tested value. All samples were preheated to 37 °C in a hot water bath and further heated up to 44 °C via a homemade inductive heating applicator (Figure 21).

The measurements were carried out in AMFs with frequency 525 kHz and amplitude of about 9 kA⋅m^−1^. After reaching 44 °C, this temperature was maintained for 30 min, which is a common treatment time for hyperthermia sessions [5]. To determine cell viability, the MTT (3-(4,5-dimethylthiazol-2-yl)-2.5-diphenyltetrazolium) test was performed. The results obtained indicate the cytotoxic effect of the hyperthermia treatment with Nanoembosil^®^. Moreover, the observed effect increases with the concentration of the mediator (Figure 22).

## 5. Concluding Remarks and Future Perspectives

Of considerable interest is the development of magnetic polymeric composites to treat malignant tumors of parenchymal organs by AEH. In this method, it is possible to achieve significant benefits from the dual effect of embolization of the tumor vascular system and subsequent MH. Embolization blocks the tumor due to the stagnation of its blood supply and leads to the shrinkage of the tumor. In turn, magnetic hyperthermia promotes the heating of the tumor to temperatures of 43–44 °C and can be repeated several times. The combined effect of embolization and hyperthermia on a tumor leads not only to ischemic necrosis but also to programmed cell death (apoptosis). Moreover, AEH combined with RT and ChT can significantly improve the survival of patients with various types of cancer.

The clinical purposes dictate the choice of materials for AEH. Thus the development of silicone-based magnetic composites that possess the set of properties necessary for conducting AEH, namely, the ability for secure embolization of the tumor blood vessels, radiopacity, and the high heating rate at moderate frequencies and amplitudes of AMFs, allowed for medical applications. The possibility of securing embolization was achieved by the optimization of the composition of the embolic agent to provide low initial viscosity (0.2–0.3 Pa·s^−1^), allowing the delivery and distribution of the material uniformly in tumor’s blood vessels, and 20–25 min induction period after which the viscosity of the composite rapidly increases forming soft embolus. Such embolus displays rubber–elastic properties with shear modulus lower than arteries, within the range of HT temperatures, and higher thermal expansion coefficient than blood. Therefore, silicone-based magnetic elastomer can deform with the blood vessels and prevent blood flow recovery during heating.

The in vivo study has shown that intra-arterial administration of Nanoembosil^®^ to animals with intramuscular developed tumors (PCI/mouse and VX2/rabbits) significantly inhibits the tumor growth rate. In vivo study of Nanoembosil^®^ should be continued towards using a combined method including embolization followed by MH. Moreover, when studying the dynamics of tumor growth, the possible influence of colloidal platinum, which is formed during the hydrosilylation reaction of hydro- and vinyl-functional silicone polymers, must be considered. In vitro and in vivo study have shown that Pt-NPs can lead to high toxicity due to their large surface area [142]. 

To increase the effectiveness of the MH, in addition to mastering the shape and size of magnetite and maghemite NPs, other ferro- and ferri-magnetic materials can be used to increase the heating efficiency in AMFs at moderate frequencies and amplitudes, such as exchange-coupled MNPs with a core–shell structure [120], as well as hybrid systems, that is, combinations of soft and hard magnets or iron oxide NPs doped with Co [143]., The transarterial embolization must be controlled by enhancing the X-ray contrast with the addition of radiopaque materials or binding radionuclides to coated MNPs [144].

The future challenge lies in developing mathematical heat transfer models in tumors and their surrounding biological tissue to preserve healthy tissue [145].

## Figures and Tables

**Figure 1 nanomaterials-11-03402-f001:**
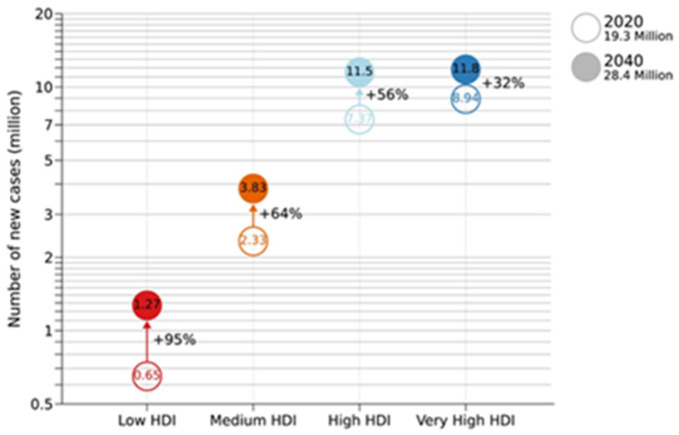
Projected number of new cancer cases in 2040 according to the 4-Tier Human Development Index. Source: GOBOCAN 2020. Reprinted with permission [1]. Copyright 2021, John Wiley, and Sons.

**Figure 2 nanomaterials-11-03402-f002:**
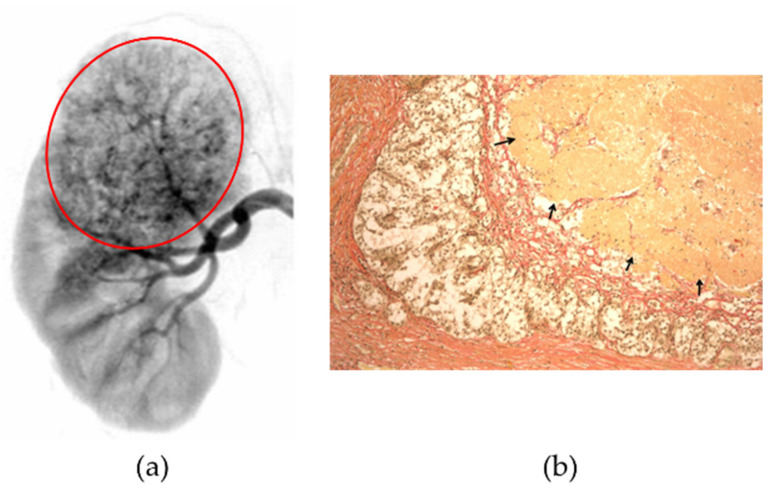
(**a**) Roentgenograph of the kidney after transarterial embolization with Ferrocomposite^®^; (**b**) Results of histological analysis indicating massive necrosis of tumor tissue of patient’s kidney after ferromagnetic embolization and capacitive RF hyperthermia [66].

**Figure 3 nanomaterials-11-03402-f003:**
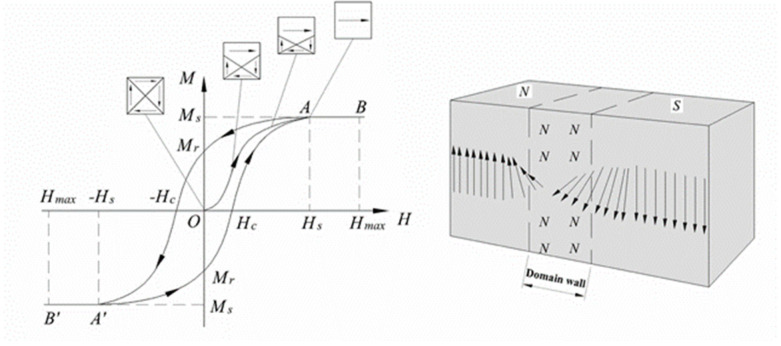
Schematic representation of initial magnetization curve, hysteresis loop, and domain wall structure for a typical ferromagnetic material, where M_S_ is the saturation magnetization, M_r_ is the remanent magnetization at H = 0, and H_C_ is the coercivity.

**Figure 4 nanomaterials-11-03402-f004:**
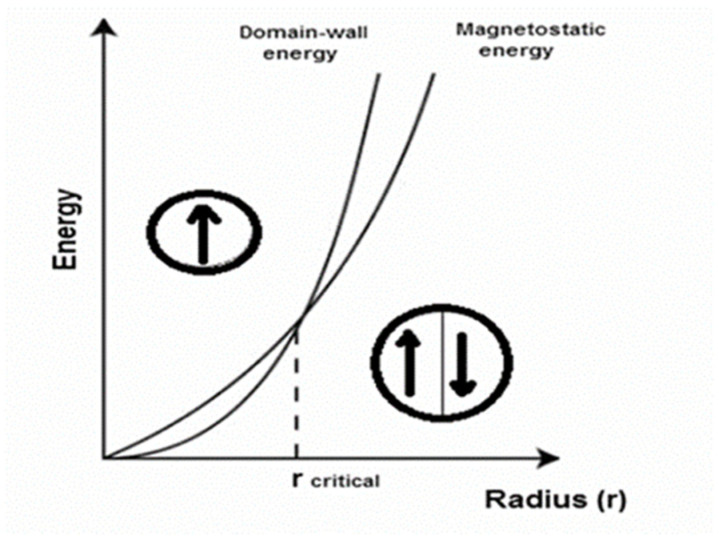
The relative stability of multidomain and single-domain particles. Reprinted with permission [78]. Copyright 2021, Cambridge University Press.

**Figure 5 nanomaterials-11-03402-f005:**
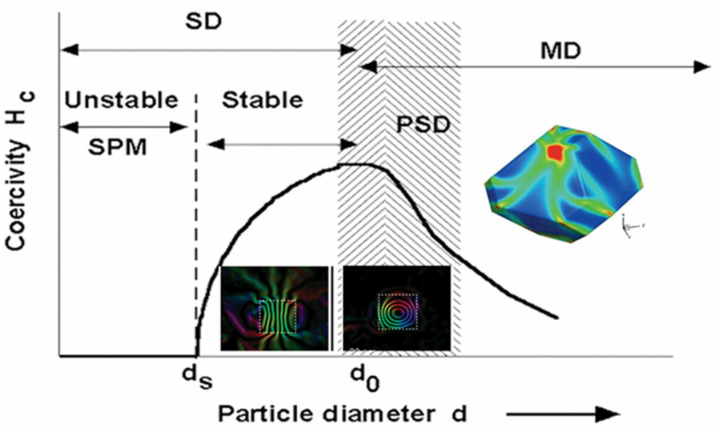
Schematic representation of the transitional state in magnets that spans the particle size range between SD and MD states. The inset collar pictures indicate the direction of the magnetic induction in magnetite particles with 25 nm, 200 nm, and microns sizes, modified from. Reprinted with permission [82]. Copyright 2021, American Chemical Society.

**Figure 6 nanomaterials-11-03402-f006:**
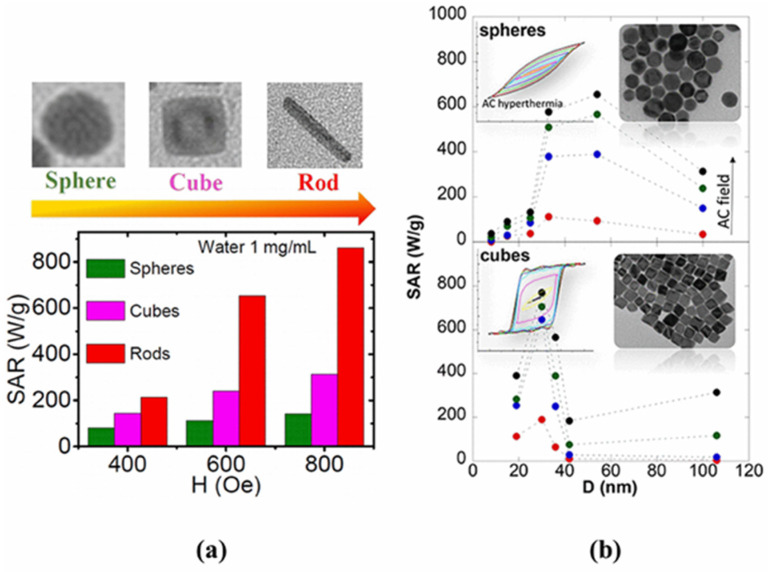
(**a**) SAR vs. H for the Fe_3_O_4_ NPs (spheres, cubes, nanorods) with the volume of about 2000 nm^3^, (**b**) SAR vs. H for the sphere and cube-shaped Fe_3_O_4_ NPs dispersed in water under AMF (300 kHz, from tens to 800 Oe). Reprinted with permission [87,88]. Copyright 2021, American Chemical Society.

**Figure 7 nanomaterials-11-03402-f007:**
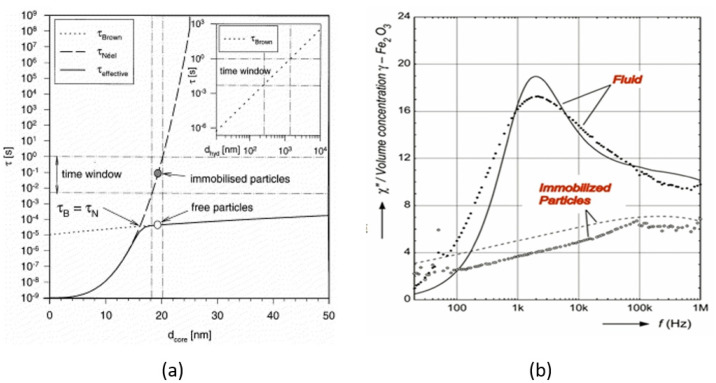
(**a**) Néel and Brown relaxation times calculated over a range of particle sizes for a water-based magnetite ferrofluid [96]; (**b**) Imaginary part of susceptibility of maghemite based aqueous suspension in comparison to the identical particles immobilized in the gel. Reprinted with permission [96,98]. Copyright 2021, Elsevier.

**Figure 8 nanomaterials-11-03402-f008:**
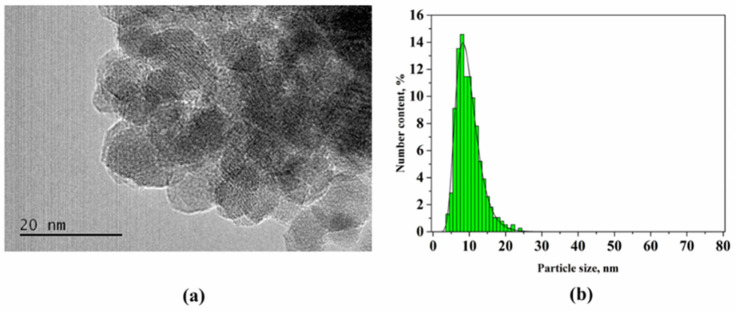
(**a**) TEM image of the five-minute reaction product for magnetite synthesized by coprecipitation method, and (**b**) particle size distribution histogram. Reprinted with permission [93]. Copyright 2021, American Chemical Society.

**Figure 9 nanomaterials-11-03402-f009:**
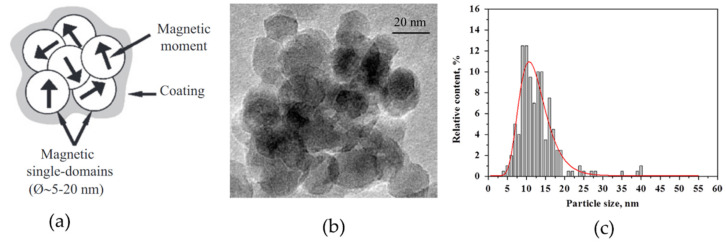
(**a**) Schematic picture of dense aggregate (multicore particle) with surface coating [106], (**b**) TEM image of magnetite NPs clustered into dense aggregate, and (**c**) particle size distribution in aggregate. Reprinted with permission [93,106]. Copyright 2021, Elsevier, and American Chemical Society.

**Figure 10 nanomaterials-11-03402-f010:**
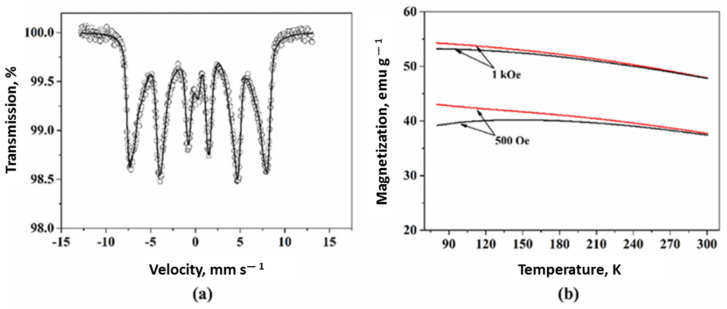
(**a**) Mössbauer spectroscopy and (**b**) FC/ZFC magnetization curves of as-prepared magnetite NPs in a powder form. Reprinted with permission [107], Copyright 2021, Elsevier.

**Figure 11 nanomaterials-11-03402-f011:**
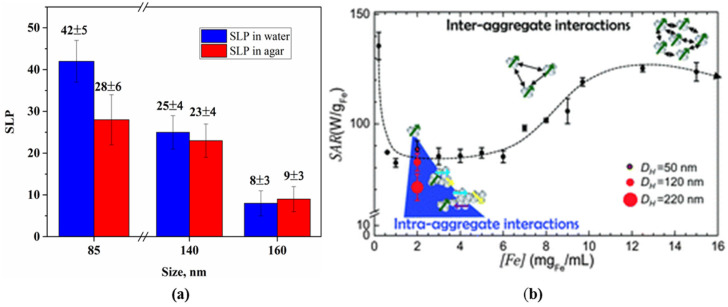
(**a**) SLP for water and agar dispersions of magnetite-based multicore particles as a function of hydrodynamic size in AMF of 1048 kHz and 5.8 kA/m [93]; (**b**) Effect of intra and intercluster magnet–dipole interactions in the dispersion of magnetite NPs on SAR as a function of concentration and hydrodynamic cluster size (D_H_) obtained under the given AMF conditions of 105 kHz and 13.1 kA/m [111]. Reprinted with permission [93,111]. Copyright 2021, Elsevier and Royal Society of Chemistry.

**Figure 12 nanomaterials-11-03402-f012:**
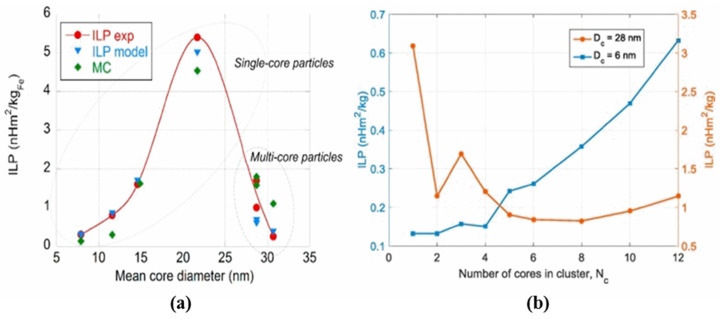
(**a**) Experimental (red) and simulated ILP data analysis for noninteracting (blue) and interacting single-core particles (green); (**b**) Simulated ILP versus the number of cores in a core cluster for two core sizes, below (6 nm) and above (28 nm) the core size maximizing the ILP value. Reprinted with permission [119]. Copyright 2021, Elsevier.

**Figure 13 nanomaterials-11-03402-f013:**
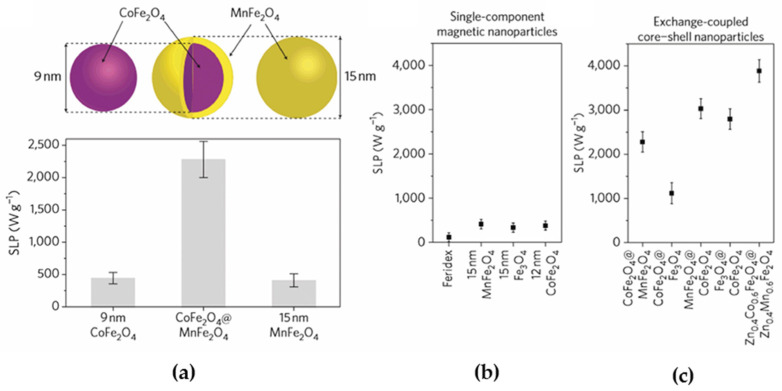
(**a**) Schematic representation of an exchange-coupled core–shell CoFe_2_O_4_@MnFe_2_O_4_ nanoparticle and its SLP value compared with SLP values of single-phase CoFe_2_O_4_ and MnFe_2_O_4_; (**b**,**c**) SLP values of various combinations of core–shell NPs and single-component NPs measured at f = 500 kHz and H = 37.3 kA/m. Reprinted with permission [120]. Copyright 2021, Springer Nature.

**Figure 14 nanomaterials-11-03402-f014:**
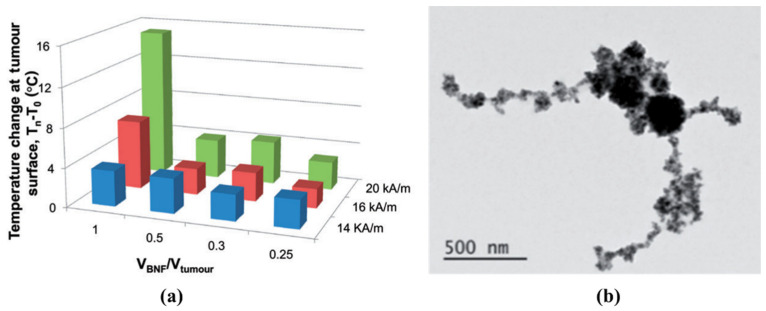
(**a**) Temperature change at the tumors of mice with hepatocellular carcinoma after injection of magnetite NPs emulsion in Lipiodol (cBNf-lip) followed by MH at 155 kHz and amplitudes within 14–28 kA/m. (**b**) TEM image of cBNf-lip. Reprinted with permission [62]. Copyright 2021, Taylor & Francis.

**Figure 15 nanomaterials-11-03402-f015:**
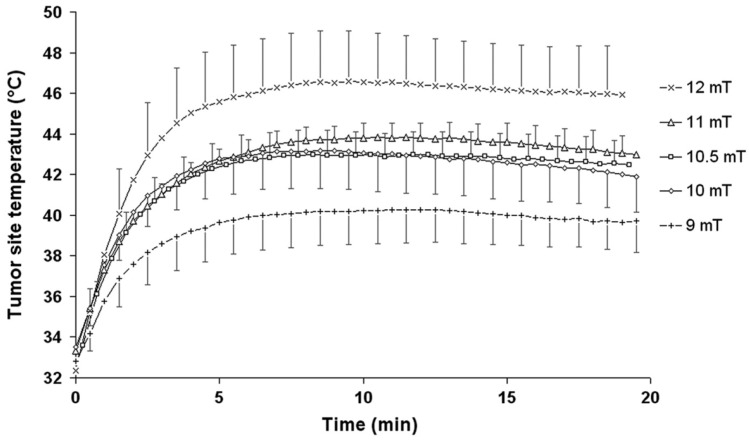
Thermogram of mean intratumoral tumor temperature as a function of treatment time and magnetic field strength. Reprinted with permission [64]. Copyright 2021, Taylor & Francis.

**Figure 16 nanomaterials-11-03402-f016:**
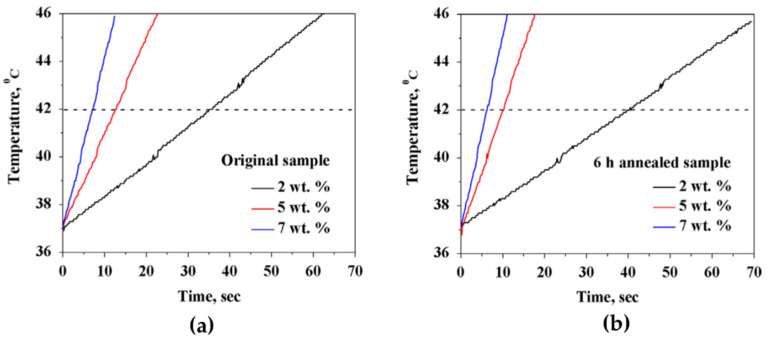
Inductive heating of magnetite (**a**) and maghemite (**b**) glycerol dispersions in AMF (525 kHz, 7.6 kA/m). Reprinted with permission [107]. Copyright 2021, Elsevier.

**Figure 17 nanomaterials-11-03402-f017:**
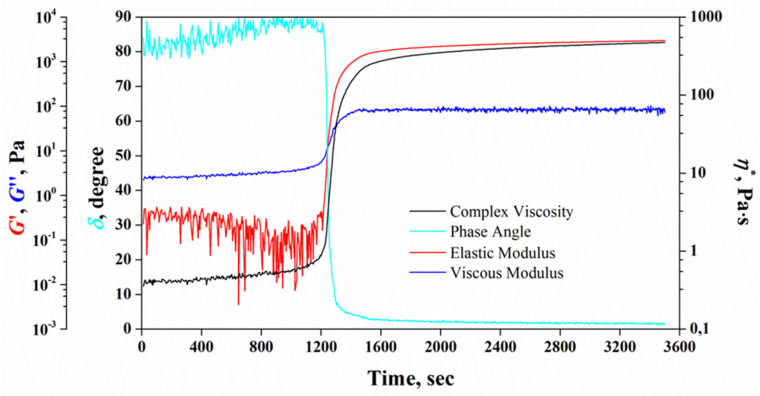
Kinetics of Nanoembosil^®^ formation. Reprinted with permission [68]. Copyright 2021, Elsevier.

**Figure 18 nanomaterials-11-03402-f018:**
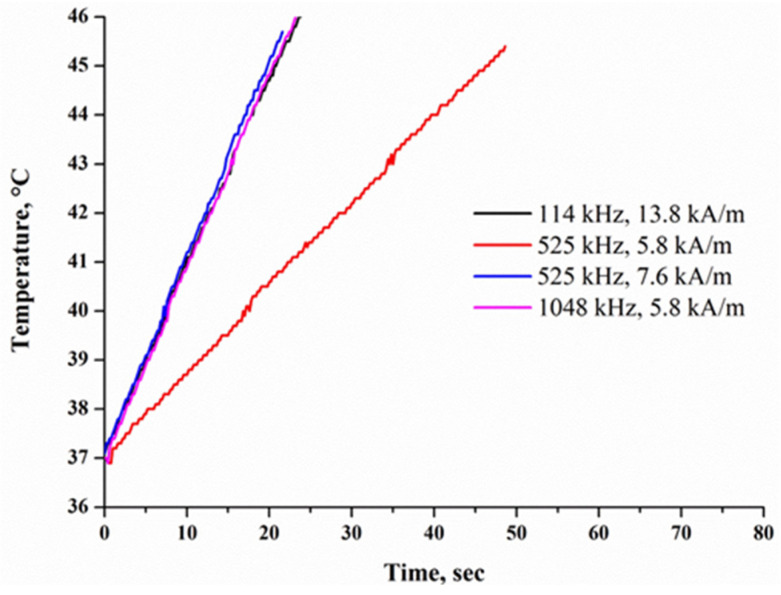
Inductive heating of composites with 7 wt.% of NPs at different alternating magnetic field parameters. Reprinted with permission [68]. Copyright 2021, Elsevier.

**Figure 19 nanomaterials-11-03402-f019:**
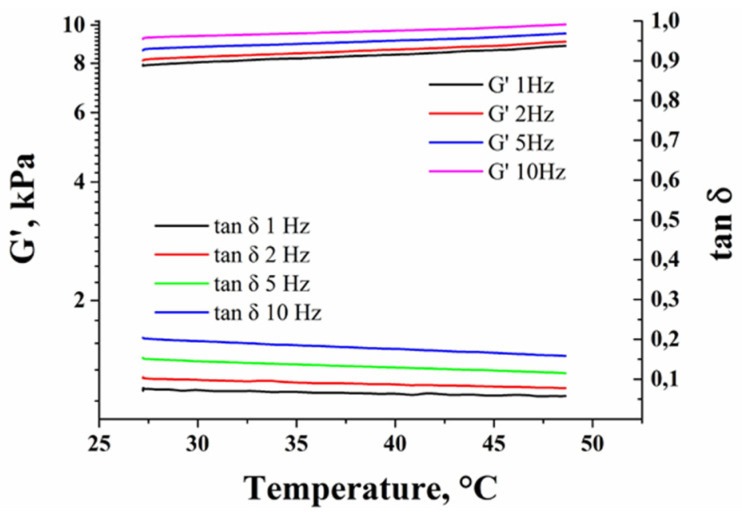
Temperature dependence of shear modulus and loss tangent at different shear rates for Nanoembosil^®^. Reprinted with permission [68]. Copyright 2021, Elsevier.

**Figure 20 nanomaterials-11-03402-f020:**
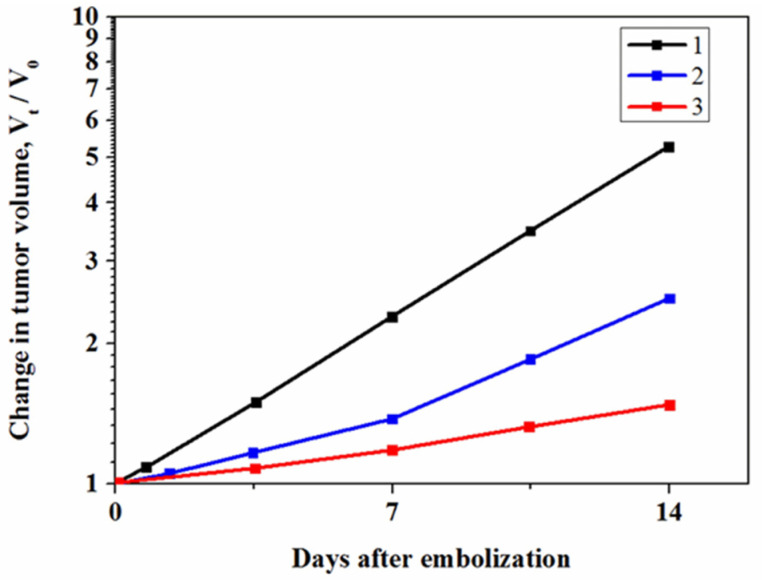
Inhibition of rat liver tumor growth after embolization of tumor vascular system by silicone-based magnetic composites: 1: test group; 2: embolization with composite I (administrative dose 0.1 mL); 3: embolization with composite II (administrative dose 0.2 mL); standard deviation is ±0.5.

**Figure 21 nanomaterials-11-03402-f021:**
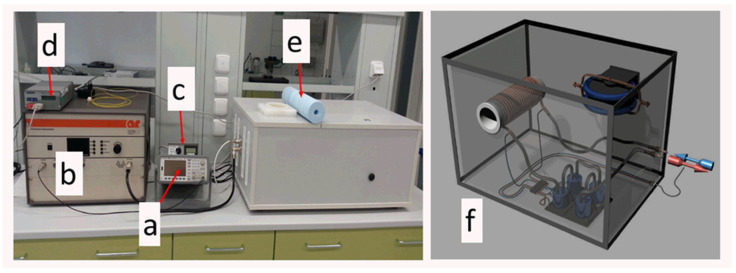
Laboratory device for measuring the heating ability of materials and conducting in vitro and in vivo studies: (**a**) signal generator, (**b**) signal amplifier, (**c**) AMF amplitude meter, (**d**) temperature recorder with fiber optic thermocouples, (**e**) sample holder, (**f**) schematic illustration of a device with a horizontal position of an impedance coil for in vivo study.

**Figure 22 nanomaterials-11-03402-f022:**
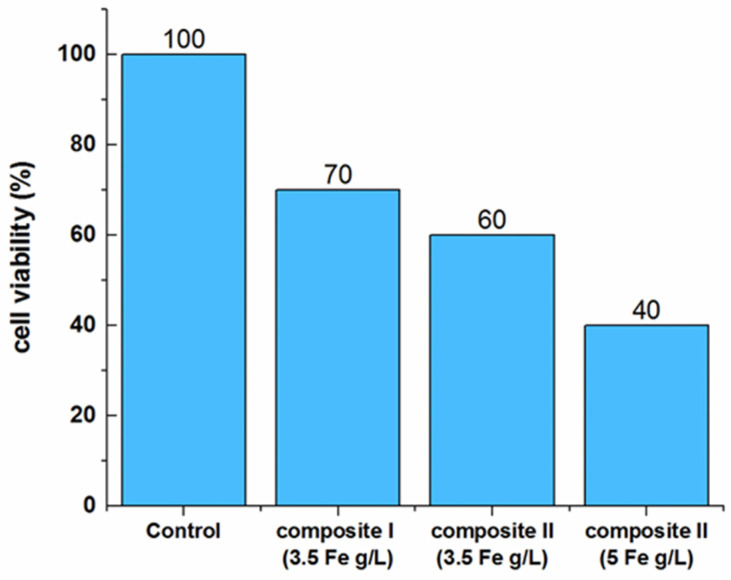
Results of in vitro test for magnetic hyperthermia-treated HepG2 cells. The AFM parameters are f = 525 kHz, H = 9 kA⋅m.

**Table 1 nanomaterials-11-03402-t001:** Magnetic phase: Iron oxide NPs Structural and magnetic properties.

Property	Magnetite	Maghemite
d_TEM_ (nm)	13	13
σ_TEM_	0.3	0.3
d_XRD_ (nm)	12	12
ε (%)	0.3	0.6
Magnetite content determined from XRD (%)	72	8
Magnetite content determined from MS (%)	60	0
M_s_ (emu g^−1^)	56 ± 2	48 ± 1
M_r_ (emu g^−1^)	0.8 ± 0.2	0.8 ± 0.2
H_c_ (Oe)	11 ± 4	10 ± 3
SLP (W/g_Fe_)	23.0 ± 0.6	20.3 ± 1.5

d_TEM_, d_XRD_—average particle size determined by TEM and XRD, respectively; σ_TEM_—polydispersity index; ε—crystal lattice strain; M_S_—Mössbauer spectra.

**Table 2 nanomaterials-11-03402-t002:** Polymer phase: Characteristics of raw polymer for Nanoembosile^®^ preparation.

Reagent	Molar Weight (g·mol^−1^)	Polydispersity	Viscosity @ 25 °C (Pa·s)	Concentration of Substitutions (wt. %)
PVS	100,126	1.62	2.6	0.1–0.4, vinyl groups
PHS	17,202	1.83	0.65	0.55, hydrosubstitutions
PDMS	73,78	1.20	0.03	-
CTS	345	-	3.9	-
Speier’s catalyst	-	-	-	Hexachloroplatinic acid [H_2_PtCl_2_]·H_2_O dissolved in PDMS
Karstedt’s catalyst	-	-	-	Platinum [0] complex containing vinyl–siloxane ligands

PVS [poly(dimethylsiloxane-co-methylvinylsiloxane)]; PHS [poly(dimethylsiloxane-co-methylhydrosiloxane)]; PDMS [poly(dimethylsiloxane)]; CTS/Cyclotetrasiloxan [1,3,5,7-tetravinyl-1,3,5,7-tetramethylcyclotetrasiloxane].

**Table 3 nanomaterials-11-03402-t003:** Nanoembosile^®^ composition.

Composite Type	Concentration of the Initial Components, wt. %							η*_in_	η*_fin_	t_in_
	PVS	CAT	PHS	PDMS	CTS	NPs	KI	(Pa/s)		(min)
I	40	2	11	33	-	7	7	0.3	3000	20
II	36	2	10	32	6	7	7	0.25	3000	25

η*_in_ is initial viscosity, η*_fin_ is final viscosity, t_in_ is the duration of the induction period.

**Table 4 nanomaterials-11-03402-t004:** SLP values (W·g_Fe_^−1^) and heating rates (°C/min) of Nanoembosil^®^ in AMF of various frequencies and amplitudes.

AC Magnetic Field	f (kHz)	114	525	1048	
	H (kA·m^−1^)	13.8	5.8	7.6	5.8
SLP		8.6 ± 0.1	4.3 ± 0.4	8.6 ± 1.0	9.0 ± 1.5
Heating rate	(°C·min^−1^)	13.8	5.8	7.6	5.8

**Table 5 nanomaterials-11-03402-t005:** The effect of embolization by silicone-based magnetic composites on the dynamics of PC1 tumor growth in rats.

Group of Animals	Dose (mL)	V_t_/V_0_ n Days after Embolization		τ_2_ (Days)	τ_trial_/τ_control_ (%)
		3	7		
Test group saline infusion	0.2	2.3	5.3	3.0	-
Group 1 Composite I	0.1	1.4	2.5	6.0	2.0
Group 2 Composite II	0.1	1.4	2.5	6.0	2.0
Group 3 Composite II	0.2	1.2	1.5	>11	4.0

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
