# Peer review of "Magnetic Nanomaterials for Arterial Embolization and Hyperthermia of Parenchymal Organs Tumors: A Review"

_nanomaterials, 2021, doi:10.3390/nano11123402_

Round 1

Reviewer 1 Report

In this manuscript the topic of magnetic hyperthermia has been reviewed and in particular, arterial embolization hyperthermia. As the use of hyperthermia alone does not allow a proper heat localization in the tumor, the combination with radiotherapy or chemotherapy is one of the most used resources in order to increase the heating power. In most of the cases the carrier medium is water or saline (Magnetic Fluid Hyperthermia), nevertheless sometimes oily contrast media, in-situ gelling materials and low-viscosity polymers can be employed (Arterial Embolization Hyperthermia). The authors focus the review in this last topic and analyze the different factors that can enhance magnetic hyperthermia: materials size, material composition and degree of crystallinity, shape, magnetic response, anisotropy, interactions between nanoparticles, core-shell structures, … Finally, they explain the characteristics of an embolic agent based on maghemite-based silicone composition (NanoembosilÒ) together with the results obtained by in vivo study. Despite the bibliographic search concerning magnetic hyperthermia I consider that the manuscript should be improved with some additional information.

  1. The heat production of magnetic nanoparticles is due to the hysteric dynamic response in an AC magnetic field, so the direct determination of the hysteresis to calculate the SAR provides a lot of information of collective phenomena (M. Bekovic, IEEE Trans. Magn. 2010; E. Garaio, Meas Sci. Technol. 2014, M. Cosson, Biochimica et Biophysica Acta (BBA), 2017; I. Andreu, Int. J. Hyperthermia, 2013). It would be interesting to add information from AC measurements for quantifying SAR and to obtain additional information for anisotropy or nanoparticles shape.
  2. Another interesting point would be to consider iron oxide nanoparticles with non-fluctuating moments to generate maximum heating power at large enough magnetic fields (I. Castellanos-Rubio, Nanoscale, 2019) as the dependence of SAR with magnetic field is not linear and depends on the size of the NPs. This fact should also be added in the manuscript when discussing the influence of size.
  3. Authors claim that in vivo experiments have been performed in Blokhin Russian Cancer Research Centre in Moscow but I could not find a proper citation of this work.
  4. A careful reading of the manuscript could avoid English misspelling of some words.

The overall quality of the article is quite enough to consider for publication after considering the above improvements.

Author Response

We would like to thank the reviewer for the careful analysis of our work, valuable comments and suggestions that helped us to put more attention on the complex of material properties that affect the heating efficiency. Below we give the answers to all the comments posed. Necessary changes were made in the text of the manuscript to fix the lacunae pointed out by the reviewer.

 Reviewer's comment:

In this manuscript the topic of magnetic hyperthermia has been reviewed and in particular, arterial embolization hyperthermia. As the use of hyperthermia alone does not allow a proper heat localization in the tumor, the combination with radiotherapy or chemotherapy is one of the most used resources in order to increase the heating power. In most of the cases the carrier medium is water or saline (Magnetic Fluid Hyperthermia), nevertheless sometimes oily contrast media, in-situ gelling materials and low-viscosity polymers can be employed (Arterial Embolization Hyperthermia). The authors focus the review in this last topic and analyze the different factors that can enhance magnetic hyperthermia: materials size, material composition and degree of crystallinity, shape, magnetic response, anisotropy, interactions between nanoparticles, core-shell structures, … Finally, they explain the characteristics of an embolic agent based on maghemite-based silicone composition (NanoembosilÒ) together with the results obtained by in vivo study. Despite the bibliographic search concerning magnetic hyperthermia I consider that the manuscript should be improved with some additional information.

The overall quality of the article is quite enough to consider for publication after considering the above improvements.

Q1. The heat production of magnetic nanoparticles is due to the hysteric dynamic response in an AC magnetic field, so the direct determination of the hysteresis to calculate the SAR provides a lot of information of collective phenomena (M. Bekovic, IEEE Trans. Magn. 2010; E. Garaio, Meas Sci. Technol. 2014, M. Cosson, Biochimica et Biophysica Acta (BBA), 2017; I. Andreu, Int. J. Hyperthermia, 2013). It would be interesting to add information from AC measurements for quantifying SAR and to obtain additional information for anisotropy or nanoparticles shape.

Answer: The key requirement in MH is maximizing of heat generation within medically safe limits of the AMF. Therefore, particle size and particle size distribution have to be taken under control. Experimental determination of heating effect of magnetic materials is usually conducted by non-adiabatic calorimetric method [I. Andreu, E. Natividad, Accuracy of available methods for quantifying the heat power generation of nanoparticles for magnetic hyperthermia. Int. J. Hyperthermia, 2013, 29(8), pp. 739-751], and rarely by adiabatic calorimetry [E. Garaio et. al, A multifrequency eletromagnetic applicator with an integrated AC magnetometer for magnetic hyperthermia experiments. Meas Sci. Technol. 2014, 25, 115702]. It is possible also to predict size-dependent heating efficiency of MNPs by stochastic Neel-Brown Langevin equation Monte-Carlo simulations [Coffey, W.T., Kalmykov, Y.P., Thermal fluctuations of magnetic nanoparticles: Fifty years after Brown. Journal of Applied Physics 112(12),121301]. K.M. Krishnan used this method to calculate the heating efficiency of MNPs with a size of 10-30 nm and various values of effective anisotropy constant (K = 4000 J/m3 -11 J/m3) and damping parameter (a=0.5-1). The magnetic parameters of MNPs were obtained from VSM. Experimental and simulated results have shown that the maximum SLP value demonstrated particles in the size range of 22-28 nm. Moreover, MC simulation reviled a strong dependent of SLP on K. [Engelmann, U.M., Shasha, C., Teeman, E., Slabu, I., Krishnan, K.M., Predicting size-dependent heating efficiency of magnetic nanoparticles from experi-ment and stochastic Néel-Brown Langevin simulation. Journal of Magnetism and Magnetic Materials, 2019, 471, pp. 450–456]. Besides, various empirical and analytical methods are used to evaluate the SLP from an experimental setup, such as the initial slope, corrected slope, Box-Lucas and steady state methods [D.E. Bordelon et al., J Appl Phys 2011]. Although recently developed bio-heat models for MH are used with a view to understand of heat transfer phenomena in living tissue. All these methods are fully considered in newly published articles of I. Raouf et al. [Thermal Biology 2020, 91, 102644] and M. Suleman et al [J Thermal Analysis and Calorimetry 2021 146:1193-1219]. Thus, we believe that there is no need to describe in detail the calculation models in our article.

Q2 Another interesting point would be to consider iron oxide nanoparticles with non-fluctuating moments to generate maximum heating power at large enough magnetic fields (I. Castellanos-Rubio, Nanoscale, 2019) as the dependence of SAR with magnetic field is not linear and depends on the size of the NPs. This fact should also be added in the manuscript when discussing the influence of size.

 Answer: It is known that MNPs with particle size larger than 20 nm are in a stable-domain state with ferromagnetic like behavior when the magnetic moment is pinned along the magnetic anisotropy axis as a result of effective magnetic anisotropy [Coey, J.MD. Ferromagnetism and exchange, Cambridge University Press 2010]. Such NPs exhibit much higher heat loss in AMF. These class of magnetic nanomaterials include also the novel octahedral monocrystalline magnetite NPs obtained by thermal decomposition [I. Castellanos-Rubio, Nanoscale, 2019]. Nevertheless, due to the octahedral morphology these NPs show one of the largest SARs rates reported to date for a colloidal suspension of magnetite. Such behavior has been explained by the shape of NPs that imprints a biaxial or bi-stable character to the magnetic anisotropy.

Q3. Authors claim that in vivo experiments have been performed in Blokhin Russian Cancer Research Centre in Moscow but I could not find a proper citation of this work. A careful reading of the manuscript could avoid English misspelling of some words.

Answer: The results of in-vivo study of NanoembosilÒ is published in the Russian journal «Problems in Oncology»: Treshalina, H.M.; Yakunina, M.N.; Makovetskay, K.N.; Stangevskiy, A.A. Dynamics of tumor growth under the action of the new nano-ferrimagnetiс nanoembosil with transarterial introduction. Problems in Oncology 2020. Ref. 139

Reviewer 2 Report

This manuscript widely reviews the potential of magnetic nanomaterials for arterial embolization hyperthermia (AEH) and its application for treatment of malignant tumors.  The review reports detailed information about the mechanisms of magnetic losses in nanomaterials, magnetostructural properties, the heating efficiency and its relation to interparticle magnetic interactions and to the role of the carrier medium.  The authors show in-vitro and in-vivo preclinical trials of magnetic nanomaterials - mainly the maghemite-based silicone composition (Nanoembosil®) - in the AEH treatment of malignant cells and tumors. The authors convinced me that Nanoembosil® is a promising agent for AEH, however, the abundant information about the composition of raw materials (Tables 1 and 2), the heating curves (Figs. 16 and 18) measured in AMF, when Table 4 contains the parameters calculated from the AMF measurements, is unnecessary to provide in a review. I should note that the structure is not appropriate. The in-vitro and in-vivo results are discussed in the point “3. Synthesis of magnetic materials for application in AEH.” Materials have to be synthetized, characterized (this would be point 3. Synthesis and charaterization … ) and then the trials (4. In vitro in vivo…) may follow.  Careful reading is required; typos (e.g., “embolozating”,  „collar” pictures, „termototolerance [”) are still in the text. The manuscript appears to be worthy of publication, but some changes need to be made to reach the expected level.

Author Response

Reviewer's comment:

This manuscript widely reviews the potential of magnetic nanomaterials for arterial embolization hyperthermia (AEH) and its application for treatment of malignant tumors.  The review reports detailed information about the mechanisms of magnetic losses in nanomaterials, magnetostructural properties, the heating efficiency and its relation to interparticle magnetic interactions and to the role of the carrier medium. The authors show in-vitro and in-vivo preclinical trials of magnetic nanomaterials - mainly the maghemite-based silicone composition (Nanoembosil®) - in the AEH treatment of malignant cells and tumors. The authors convinced me that Nanoembosil® is a promising agent for AEH, however, the abundant information about the composition of raw materials (Tables 1 and 2), the heating curves (Figs. 16 and 18) measured in AMF, when Table 4 contains the parameters calculated from the AMF measurements, is unnecessary to provide in a review. I should note that the structure is not appropriate. The in-vitro and in-vivo results are discussed in the point “3. Synthesis of magnetic materials for application in AEH.” Materials have to be synthetized, characterized (this would be point 3. Synthesis and charaterization … ) and then the trials (4. In vitro in vivo…) may follow.  Careful reading is required; typos (e.g., “embolozating”,  „collar” pictures, „termototolerance [”) are still in the text. The manuscript appears to be worthy of publication, but some changes need to be made to reach the expected level.

Answer: Thank you for the reviewer for the suggestion regarding the organization of paragraph were synthesis of Nanoembosil® and its in-vitro and in-vivo investigation is discussed. We reorganized this part, however kept Tables 1 and 2 since it is important for correlation of the difference in embolic agents composition with results of in-vivo study. We also checked for typos and made the necessary corrections.

Reviewer 3 Report

Reviewer Recommendation and Comments for Manuscript Number 1484395 submitted for publication in Nanomaterials.

The manuscript entitled " Magnetic nanomaterials for arterial embolization and hyperthermia of parenchymal organs tumours: a review" by Kazantseva and colleagues is timely and exceptionally well written. The authors discuss the properties and synthesis of magnetic materials for their application in vitro and in vivo models for cancer treatment using AEH.

Although the authors talk about clinical trials for HT in combination with RT and ChT, and AEH, they do not analyze those trials in detail. It would be interesting to cite them and show a table with the clinical trials.

The section concluding remarks and future perspectives is too short. A little bit more discussion would be appreciated.

Author Response

Reviewer's comment:

The manuscript entitled "Magnetic nanomaterials for arterial embolization and hyperthermia of parenchymal organs tumours: A review" by Kazantseva and colleagues is timely and exceptionally well written. The authors discuss the properties and synthesis of magnetic materials for their application in vitro and in vivo models for cancer treatment using AEH.

Although the authors talk about clinical trials for HT in combination with RT and ChT, and AEH, they do not analyze those trials in detail. It would be interesting to cite them and show a table with the clinical trials.

The section concluding remarks and future perspectives is too short. A little bit more discussion would be appreciated.

Answer: We agree with reviewer that the analysis of clinical treals of HT in combination with RT and ChT is interesting and important topic since the 19.3 million new cancer cases were diagnosed worldwide with almost 10 million deaths from cancer in 2020. However, this analyze have to be done for all methods based on hyperthermia treatment, i.e. therapeutic hyperthermia, magnetic hyperthermia, arterial embolization hyperthermia etc. Such an analysis requires careful comparison of not only clinical results but also treatment techniques, which can differ significantly. This could be a topic for a new article.

The section concluding remarks and future perspectives has been expanded.